# How Does Physical Activity Modulate Hormone Responses?

**DOI:** 10.3390/biom14111418

**Published:** 2024-11-07

**Authors:** Cristina Mennitti, Gabriele Farina, Antonio Imperatore, Giulia De Fonzo, Alessandro Gentile, Evelina La Civita, Gianluigi Carbone, Rosa Redenta De Simone, Maria Rosaria Di Iorio, Nadia Tinto, Giulia Frisso, Valeria D’Argenio, Barbara Lombardo, Daniela Terracciano, Clara Crescioli, Olga Scudiero

**Affiliations:** 1Department of Molecular Medicine and Medical Biotechnologies, Federico II University, Via Sergio Pansini 5, 80131 Napoli, Italy; cristinamennitti@libero.it (C.M.); impeantonio@gmail.com (A.I.); giulia.defonzo@gmail.com (G.D.F.); alexgenti98@libero.it (A.G.); desimoner@ceinge.unina.it (R.R.D.S.); nadia.tinto@unina.it (N.T.); gfrisso@unina.it (G.F.); barbara.lombardo@unina.it (B.L.); olga.scudiero@unina.it (O.S.); 2Department of Human Exercise and Health Sciences, University of Rome “Foro Italico” Piazza L. de Bosis 6, 00135 Rome, Italy; g.farina2@studenti.uniroma4.it; 3Department of Translational Medical Sciences, University of Naples Federico II, 80138 Naples, Italy; eva.lacivita@gmail.com (E.L.C.); ginaluigi.carbone.91@gmail.com (G.C.); daniela.terracciano@unina.it (D.T.); 4CEINGE-Biotecnologie Avanzate Franco Salvatore, Via G. Salvatore 486, 80145 Napoli, Italy; diiorio@ceinge.unina.it (M.R.D.I.); dargenio@ceinge.unina.it (V.D.); 5Department of Human Sciences and Quality of Life Promotion, San Raffaele Open University, 00166 Rome, Italy; 6Task Force on Microbiome Studies, University of Naples Federico II, 80100 Naples, Italy

**Keywords:** athletes, physical activity, cortisol, testosterone, growth hormone, thyroid, insulin, catecholamines

## Abstract

Physical activity highly impacts the neuroendocrine system and hormonal secretion. Numerous variables, both those related to the individual, including genetics, age, sex, biological rhythms, nutritional status, level of training, intake of drugs or supplements, and previous or current pathologies, and those related to the physical activity in terms of type, intensity, and duration of exercise, or environmental conditions can shape the hormonal response to physical exercise. The aim of this review is to provide an overview of the effects of physical exercise on hormonal levels in the human body, focusing on changes in concentrations of hormones such as cortisol, testosterone, and insulin in response to different types and intensities of physical activity. Regular monitoring of hormonal responses in athletes could be a potential tool to design individual training programs and prevent overtraining syndrome.

## 1. Introduction

It is widely acknowledged that elite athletes exhibit changes in their hormonal composition due to conditioning; that is the adaptive response to training to become physically fit [1]. The influence of physical training on the hormonal system of athletes is intricate. Various factors, including the intensity and length of training, diet and energy levels, gender, sex, age, and stage of sexual development, all play a role in shaping the hypothalamus and pituitary gland responses to physical strain [2]. In particular, the intensity and duration of training stimulate the hypothalamic–pituitary–adrenal (HPA) axis, leading to increased cortisol levels, while an inadequate diet and low energy availability suppress reproductive hormones [3,4]. Gender and sex differences modulate the hormonal response, with variations related to testosterone and estrogen levels [5]. Age affects the reactivity of the hypothalamic–pituitary axis, with an attenuated response observed in older adults compared to younger individuals [6]. Additionally, stages of sexual development, such as puberty and menopause, significantly alter the endocrine response, highlighting the importance of a personalized approach to training [7].

To clarify the role of exercise on the endocrine system, it is imperative to delineate between the acute effects of a single exercise bout on hormonal dynamics and the enduring impacts of sustained exercise training on hormonal profiles. In the acute context, virtually all modalities of physical activity elicit alterations in the circulating concentrations of hormones, typically resulting in increased levels, often corresponding to the intensity and duration of the exercise endeavor. Acute responses of hormones to physical activity depend on exercise variables, such as intensity, volume, rest intervals, repetition velocity, frequency, and exercise selection and sequence [8,9]. Manipulation of these variables leads to an optimal neuroendocrine response, the system of primary importance in acute exercise performance, and in consecutive tissue remodeling [8]. Instead, in addition to these and previously described variables, exercise training adaptations are connected to program design, including concepts like progressive overload, variation, and specificity [9]. Although changes in the mode of exercise can engender variability in the magnitude of these responses, when assessing the enduring repercussions of exercise, evaluation must extend to resting (basal) hormonal states and the responses to subsequent exercise bouts following a period of training [10,11,12].

After prolonged exercise training, basal hormone levels typically show specific changes, experiencing either a marginal increase or a slight reduction. Table 1 summarizes gender-specific hormonal variations: the first column presents the baseline levels of various hormones, specifying the gender in which they are more predominant. The last column shows whether these gender differences have increased or leveled off. The intermediate columns report the hormonal variations observed during regular training compared to acute physical exercise [13,14,15,16,17,18,19,20,21,22,23]. This phenomenon is notably influenced by the “basement effect”, wherein detectable reductions in hormone levels are constrained, particularly when approaching values near zero [24]. Consequently, substantial further reductions become challenging to discern.

Upon engaging in acute exercise sessions after chronic training regimens, many hormonal responses are attenuated compared to those observed before the initiation of training interventions, even maintaining the same directional trend. These diminished responses are often attributable to reduced stress reactivity during exercise bouts and enhanced sensitivity of target tissues, reflecting adaptations to the training stimulus [25].

A framework for understanding how the body’s hormonal responses are regulated during exercise is the hormonal exercise response model (HERM). It describes these responses in three phases: initially, exercise triggers rapid sympathetic nervous system activation, releasing catecholamines and altering insulin and glucagon levels. As exercise continues, the hypothalamus stimulates the pituitary gland, which releases hormones like cortisol. In prolonged exercise, additional hormones from the pituitary and peripheral glands are involved, alongside fluid regulation and cytokine release from muscles. The model illustrates how these responses evolve from neural- to feedback-driven mechanisms as exercise duration increases [25].

In summary, these principles underscore the interplay between acute and chronic exercise in modulating endocrine dynamics, particularly concerning reproductive hormones [26].

Standardizing experimental conditions is essential but challenging, particularly in team sports. Consequently, comparing studies and obtaining sufficient data for each sport or athlete can be difficult, leading to discrepancies. Nonetheless, general patterns of hormonal response can be identified.

GH (growth hormone) and IGF-1 (insulin-like growth factor-1) are key players in promoting protein synthesis and maintaining body muscle mass [27]. They are involved in muscle processes like differentiation, hypertrophy, and angiogenesis. GH regulates circulating IGF-1 levels, mostly synthesized by the liver [28]. After puberty, GH secretion and circulating IGF-1 levels decline. GH release is potently stimulated by physical exercise, with resistance exercises triggering a greater response than sprint and endurance exercises [29]. Circulating IGF-1 levels may increase in response to various types of physical training. However, increasing IGF-1 levels after acute exercise is not necessarily related to exercise-induced GH secretion.

Cortisol, a key substance in our body, regulates many of the changes that occur in muscles when we exercise. Its amount in our blood depends on intensity and duration of exercise, level of fitness, diet, and circadian rhythm. These factors affect cortisol basal/resting level, as well. Although the effects of sports on cortisol levels are well described, some studies have failed to find variation in salivary cortisol values [30], while others have noted a decrease or increase in its presence in the blood [31]. In particular, some studies have highlighted the association of a significant increase in cortisol with better performance in soccer, especially in intensive training programs [32]. Nikolovski et al. examined variations in cortisol levels between training and matches, finding an increase during competitions due to higher physical and psychological stress, despite training potentially involving a greater physical load. Intense physical contact and explosive actions during matches contribute to a higher physical and psychological load, influencing cortisol responses. Therefore, cortisol levels are determined by various factors, including the competition context, playing time, number of contacts, and psychological pressure [33,34]. Cortisol is often used as an indicator of excessive overtraining and stress in athletes, which can lead to a reduction in performance [35]. Overtraining syndrome (OTS) is an imbalance between training and recovery, exercise and exercise capacity, and stress and stress tolerance [2]. Accordingly, cortisol has a catabolic effect on muscle tissues, promoting muscle fiber breakdown, and can also reduce the production of anabolic hormones, which promote muscle growth [36] (Figure 1).

A reduction in the total triiodothyroxine (TT3), an indicator of energy status, signals minor activity of the hypothalamic–pituitary–thyroid (HPT) axis [37]. This axis is often suppressed in individuals of African ancestry (AAs) compared to those of European ancestry (EAs) and sedentary women. While levels of TSH and total and free thyroxine (T4) generally remain normal, a decline in TT3 concentration is linked to a lower resting metabolic rate and subsequently lower leptin levels [37,38].

Gonadal and adrenal steroids are often used as biomarkers to detect overtraining in athletes [39]. Testosterone indicates anabolic activity, while cortisol reflects catabolic processes [40]. Increased cortisol levels due to chronic exercise stress can disrupt the body’s natural rhythms and potentially lead to health issues, like cancer and obesity [41]. Elite athletes may have varying levels of gonadal steroids based on gender and sport, with men typically exhibiting lower testosterone levels than women [42]. Female athletes may experience menstrual disorders due to overtraining or inadequate, diet but increasing energy intake can help restore normal menstruation [43,44]. Dehydroepiandrosterone (DHEA) levels are associated with muscular activity during exercise [45]. Female athletes in endurance sports may have lower testosterone and DHEAS levels compared to those engaged in strength and speed sports [46,47]. Factors like oral contraceptives, congenital adrenal hyperplasia, and menstrual disorders can affect androgen levels [48].

The aim of this review is to overview the effects of different types and intensities of physical activity on hormonal levels in the human body in order to highlight the main variations. In particular, this review has analyzed the following axes and systems: (i) GH–IGF1 axis; (ii) adrenergic system; (iii) hypothalamic–pituitary–adrenal axis; (iv) hypothalamic–pituitary–thyroid axis; (v) hypothalamic–pituitary–gonadal axis; (vi) insulin secretion.

## 2. GH–IGF1 Axis

The capacity to perform exercise strictly depends on metabolic fuel combustion, to obtain kinetic energy from chemical energy. Short-term high-intensity activity mainly requires glucose as a substrate, whereas free fatty acids (FFAs) derived from the circulation or from triglycerides stored in muscle or adipose tissue are more important for prolonged activity [49]. Oxygen (O_2_) is fundamental for the correct performance of physical exercise; O_2_ arrival to muscles depends upon adequate ventilation, transport by hemoglobin, and systemic distribution through an adequate cardiac input. GH is documented to improve exercise performance, causing an increase in the delivery of substrate and oxygen to exercising muscle, fat oxidation, and muscle strength, efficiently acting on insulin resistance [50], body composition, and thermoregulation [51,52].

Therefore, considering the anabolic and lipolytic effects of GH and the observation that an increase in GH associated with exercise precedes greater availability of metabolites, GH emerges to play an important metabolic role during exercise [53,54]. GH level increase starts from 10 to 20 min after the onset of exercise, with a peak either at the end or shortly after exercise and remains elevated up to 2 h post-exercise [55]. The variations in GH levels following acute physical exercise positively correlate with the duration and particularly with the intensity of the exercise; it is greater with resistance exercise and is influenced by the type of required muscle response. It is also conditioned by age, sex, and body composition. However, the effects of training on GH secretion are still controversial [56].

Certain GH actions take place by means of other growth factors, called somatomedins; the principal one is represented by somatomedin-C or insulin-like growth factor-1 (IGF-1) [57]. In the training cycle, the GH/IGF-1 axis behaves in a biphasic mode, showing a catabolic and anabolic phase: the first is characterized by a decrease in hormone concentration lasting 3 to 5 weeks, and the latter is characterized by an increase in hormone concentrations, after 5/6 weeks of training [58,59].

IGF-1 is a polypeptide hormone with a similar structure to proinsulin, and it mediates many of the somatic effects of GH. It is synthesized in the liver and secreted into the blood, where it circulates as complexes associated with specific binding proteins (IGFBPs) [60]. Most IGF-1 (80%) occurs in a ternary 150 kDa complex, where the insulin-like growth factor binding protein-3 (IGFBP-3) is located; less than 1% of IGF-1 is free [61,62,63]. The insulin-like growth factor binding proteins (IGFBPs) play a key role since they prolong IGF’s half-life and carrier IGF in the circulation, regulating their biological actions in an autocrine and/or paracrine fashion [64,65]. Indeed, IGFBP-3, present in many tissues, is considered the major IGF-1 carrier [65]. Several studies have shown that intensive training stimulates circulating IGF-1 (Figure 1) and IGFBP-3 because positive and significant correlations exist between these factors and physical fitness [61,66,67,68,69]. One of these investigations was conducted on 11 international rugby players, who gave their consent to participate and agreed to be sampled [70]. The study was to examine whether changes in IGF-1 and IGFBP-3 were associated with overtraining, a negative condition that was estimated by using the overtraining questionnaire of the French Society of Sport Medicine (SFMS) [71]. The results show correlation between IGF-1, IGFBP-3, and overtraining, and, more remarkably, the more tired subjects presented a fall in IGFBP-3 (Figure 1) after the match compared to their values at rest. A low level of IGFBP-3 likely exerts a protective mechanism against catabolism by increasing free IGF-1 fraction. IGFBP-3 proteolysis consequent to exercise might contribute to exercise-promoted anabolic effects [72].

Indeed, a higher IGF-1/IGFBP-3 ratio leads to greater IGF-1 availability [73]. Thus, a fall of IGFBP-3 in response to an intense bout of exercise may represent an index of tiredness in athletes [70].

Recent studies have documented that microRNA (miRNA), which are considered as gene expression regulators, can control the concentration of growth factors in body fluids through the regulation of the IGF-1/phosphatidylinositol-3-kinase (PI3K)/Ak strain transforming (AKT)/mammalian target of the rapamycin (mTOR) signaling pathway, which affects cardiac and skeletal muscle adaptation to exercise [74]. The study evaluated the expression of IGFBP-3 and of miRNAs (miRNA-124, miRNA-210, and miRNA-375) in three groups: a group of endurance athletes, a group of resistance-training athletes, and a control group, performing low physical activity [60,75]. The level of miR-124-3p, physiologically associated with bone formation and turnover, remained undetectable, while the expression of miR-210, which influences mitochondrial metabolism, cell apoptosis, and erythropoiesis regulation, was found to be higher in the resistance-training athletes. Interestingly, the higher expression level of miR-375 with a decrease in insulin levels was found in both groups of athletes, in line with the observation that this miRNA takes part in insulin secretion regulation and glucose homeostasis. This result strengthened the association of the role of miRNA-375 in glucose homeostasis with the metabolic adaptation process to physical effort [60].

## 3. Adrenergic System and Stress

Stress is well known to be a primary factor to trigger catecholamine secretion in normal conditions [76]. Catecholamine plasma concentration increases in response to various stressors, including physical exercise [76,77,78,79,80,81,82,83]. Noradrenaline synthesis occurs at the endings of sympathetic nerve fibers, and both adrenaline and noradrenaline occur in chromaffin cells of the adrenal medulla [84,85]. Whereas noradrenaline is considered both a neurotransmitter and a hormone, adrenaline is solely regarded as a hormone. Catecholamines act by utilizing membrane receptors, particularly adrenergic receptors (α and β subtypes) [86].

Catecholamines affect physical performance, e.g., in response to maximal or supramaximal exercise, by regulating glycogenolysis in skeletal muscle [85] and glycogenolysis in the liver [87,88,89,90] (Figure 1). Therefore, competitive sports take advantage of a high capacity to secrete these hormones [76,79,91] (Figure 1). Physical exercise can be considered a stress factor capable of stimulating the sympathoadrenal system [92]. The catecholamine response largely depends on the type, duration, and intensity of exercise.

As the activation of sympathoadrenal activity occurs only after an intense effort, as shown by Refs. [76,93,94], several studies have reported on adrenaline/noradrenaline concentrations and different exercise modalities. Although the mechanisms underlying these processes are not fully elucidated, there are several hypotheses to explain the association between adrenaline/noradrenaline plasma increase and exercise intensity. Combined changes in catecholamine clearance and/or the secretion rate might underlie the elevated values observed after long-duration exercise [95,96,97]. Considering the rapid adrenaline and noradrenaline increase, albeit the following decrease in their concentration, it is possible that stimulation of secretory mechanisms significantly contributes to these variations, according to Kjær et al. [98]. These authors reported a clearance reduction in various workloads, especially with a plasma adrenaline concentration > 10 times over baseline values. However, catecholamine clearance change related to supramaximal exercise is still to be clarified. According to Sacca et al., the increase in clearance during moderate exercise likely relies on the blood flow increase within the sites responsible for catecholamine elimination [99]. The fall in catecholamine clearance during intense exercise reflects a decrease in adrenaline elimination throughout the body. Therefore, hypothesizing that catecholamine clearance is likely the sum of elimination from all tissues, it could be suggested that the drop in clearance level mirrors the changes in blood mass distribution. Furthermore, catecholamine inactivation occurs at a significant level in some tissues, e.g., the liver, or at a weak rate in others, like the skin. Some authors reported a significant decrease in adrenaline clearance during moderate exercise [100]. To date, both clearance and the secretion rate depends on perfusion of large amounts of adrenaline; the increase in plasma noradrenaline seems to be associated with decreased elimination from the bloodstream [101]. However, Hagberg et al. [102], in studies of adrenaline kinetics during recovery, demonstrated that decreased rates are not due to changes in hormone elimination. Additionally, the blood sampling site may also influence the clearance value, causing a reduction in hormonal elimination in the area of interest [98].

Altogether, adrenaline and noradrenaline plasma concentrations observed during intense training sessions are likely to reflect higher secretion rates rather than a decrease in their elimination or clearance [79,80,81]. The high concentrations of catecholamines seen after short and intense exercises are primarily explained by increased secretion. Accordingly, a secretory mechanism seems mainly responsible for the higher adrenaline and noradrenaline levels after supramaximal exercise based on studies of the adrenaline index and noradrenaline elimination [91,102,103,104].

Finally, the importance of feedback mechanisms in hormonal response to physical exercise should not be neglected. Some local factors, e.g., osmolarity [105], may influence the level of muscular activity, which, in turn, might participate in regulating sympathoadrenal activity during supramaximal exercise, according to Brooks et al. [106]. However, the very rapid changes in catecholamine concentration observed at the end of this type of exercise are unlikely to be explained only by this mechanism. It is likely that alongside skeletal muscle activation, direct stimulation of brain motor centers (central command) takes part in the sympathoadrenal response to intense physical exercise. In addition, the intensity of the “upcoming” exercise may influence resting adrenaline concentrations. Kraemer et al. [107] observed that exercise intensity increases resting plasma adrenaline concentrations. A significant rise in adrenaline just before intense exercise has been reported only in trained subjects by Zouhal et al. [79] and Kraemer et al. [108].

Although physical exercise is undeniably recognized to increase adrenaline and noradrenaline concentrations in both women and men [109,110], the increased magnitude is still debated. Mainly, there are two types of studies: (i) those which do not mention sex-related differences and (ii) those which report significantly higher catecholamines in men vs. women. Many studies [109,110,111,112] have found no differences in catecholamine concentrations between men and women, either untrained or physically active, in response to submaximal effort. Similar results have been reported by Friedmann and Kindermann [113], regardless of training levels. In fact, no difference in catecholamine concentration kinetics emerged between untrained or resistance-trained men and women during a 10–17 km treadmill test (75% or 80% of their VO_2_max). Similarly, Friedlander et al. [114] did not find any difference in catecholamine responses between untrained men and women after 1 h of cycling (65% of their VO_2_max). After more intense exercise (Wingate test), Zouhal et al. [115] reported no sex-related differences in adrenaline and noradrenaline levels in highly resistance-trained men and women, matched for their VO_2_max level (correlated with lean body mass) and competition level. Consistent with these observations, Pullinen et al. found no differences between physically active men and women in response to very intense leg flexion/extension exercises, increasing intensity up to exhaustion [116].

Conversely, other investigators have reported sex-dependent responses to physical exercise in terms of catecholamine increases. Brooks et al. observed significantly higher plasma adrenaline concentrations in physically active men compared to women after a series of ten 6 s sprints on a non-motorized treadmill [106]. Similarly, in sprinting-trained men and women, matched according to their competition level, Gratas-Delamarche et al. reported adrenaline concentrations twice as high in men vs. women in response to the Wingate test [102]. Twelve-week resistance training (1 h/day cycling at 75% of VO_2_max, 5 days/week), evocated significantly higher adrenaline and noradrenaline concentrations in men after 1 h of cycling at 65% of VO_2_max, as shown by Friedlander et al. [114]. In untrained men and women, Horton et al. [117] reported higher catecholamine concentrations in men during a low-intensity (40% of VO_2_max) but long-duration (2 h) test. Carter et al. [118] observed significantly higher adrenaline concentrations in men vs. women after a resistance training program, at the end of prolonged moderate exercise (90 min at 60% of VO_2_max). At variance, Lehmann et al. [119] found much higher adrenaline and noradrenaline levels in women when comparing the catecholamine responses in males and females with the same maximal aerobic speed (MAS) at the end of an incremental treadmill exercise to exhaustion.

Many of these divergences can be explained, at least in part, by the diversity of protocols used in the studies, as emphasized by Viru [55]. Indeed, all the studies focusing on sex-dependent catecholamine responses to exercise have been performed in subjects of different ages, with different levels of physical fitness and different sports specialties. In addition, differences in blood sampling times, assay methods, exercise modality, and significantly variable intensities and durations of exercise are responsible for possible bias.

Additionally, a subject’s physical training levels and state, as well as warm-up preceding exercise must be reported in investigations. To date, the menstrual cycle phase is rarely specified in women. Of note, most of these factors can influence the plasma catecholamine level both at rest and during physical exercise and, therefore, cannot be neglected.

## 4. Hypothalamic–Pituitary–Adrenal Axis: The Play of Cortisol

Cortisol is the primary hormone of stress. It belongs to the glucocorticoid family, which makes it a steroid, produced and released by the zona fasciculata of the adrenal cortex. This hormone plays an important role in controlling blood glucose and metabolism during physical activity [120]. Cortisol is functional for training, as it promotes the production of glucose, useful as an immediate energy source during physical activity or in moments of stress, starting from proteins extracted from muscles. Additionally, it helps retain the fluids needed to protect the joints. Following an intense session of physical exercise, blood cortisol levels vary significantly due to activation of the HPA axis in response to increased physical and metabolic stress [120]. The extent of these variations is proportional to the intensity and duration of the exercise performed and to the type of exercise, being greater for anaerobic types. Additional variables include nutritional status, altitude, and the time of day, because of the circadian rhythm [121]. In fact, previous studies have measured increases in cortisol levels following acute exercise [122,123].

The confluence of an excessive training load coupled with inadequate recovery can precipitate a decline in sport-specific performance, which requires several weeks or even months to resolve, attributed to fatigue, termed OTS [124]. Improper recovery protocols, insufficient caloric intake, social stressors, and overly aggressive training regimens are recognized as key triggers of OTS and its associated states. The incapacity to sufficiently recuperate from demanding energy expenditures results in a broad dysfunctional adaptation, marked by abnormal responses across multiple physiological markers. Since OTS constitutes a diagnosis of exclusion, screening for inflammatory, metabolic, hormonal, psychiatric, and infectious etiologies is necessary because these conditions may contribute to decreased athletic performance [125,126]. While several proposed biomarkers for OTS diagnosis exist, impaired hormonal responses to maximal exercise-induced stress tests have been noted [123,127,128,129,130].

Chronic stress exposure is posited as a plausible determinant of the observed impaired hormonal responses, potentially stemming from diminished reactivity within the HPA axis. Additionally, alterations within the HPA axis, such as a blunted cortisol awakening response (CAR) and perturbed salivary cortisol patterns or axis hyper-response (Figure 1) [2], have shown promise in aiding OTS diagnosis, highlighting their potential utility in clinical practice [131,132,133,134,135]. For this reason, it is crucial to ensure that cortisol levels are always optimal to modulate physical exercise and maintain an optimal state of health. Numerous studies monitoring the levels of this hormone used a diagnostic test based on salivary sampling at multiple time points during training to visualize the trend.

Cadegiani et al. [2] evaluated the salivary cortisol rhythm (SCR) as a possible marker of fatigue. Salivary samples were collected by the athletes themselves using a specific laboratory kit at different times of the day: in the morning upon waking, 30 min after waking, at 4:00 PM, and at 11:00 PM. Other hormones of interest in this study were all evaluated through electrochemiluminescence assays. These samplings and analyses led to observing a significant difference in salivary cortisol values collected 30 min after waking. This result suggests the possibility of using this measurement as a marker for OTS.

Hough et al. started from the hypothesis that salivary cortisol levels might increase later compared to their respective plasma levels because hormones diffuse more slowly into saliva [136]. The participants were well-trained athletes undergoing an intensive training protocol, including high-intensity cycling and resistance exercise. The athletes consumed water ad libitum during the main trials, but they were not allowed to drink 10 min prior to saliva collection to avoid diluting the sample. Saliva samples were collected at 0 (pre-exercise), 10, 20, 30, 40, 50, and 60 min post-exercise in all the trials, with a minimum collection time of 2 min per subject to ensure an adequate sample volume. Finally, the saliva samples were immediately aliquoted and stored at −20 °C until analysis, which was performed using commercially available ELISA kits (Salimetrics, State College, PA, USA). The initial hypothesis was confirmed, as peak concentrations were observed at 10–20 min post-exercise for plasma cortisol and approximately 30 min post-exercise for salivary cortisol [136].

Honceriu et al. [137] conducted evaluations of both serum and salivary cortisol in professional soccer players, aiming to assess the possibility of definitively replacing serum cortisol sampling with salivary cortisol sampling due to its lesser invasiveness and ease of determination. Blood samples were collected at rest (T0), immediately after performing a cardiopulmonary exercise test (CPET) (T1), 10 min after CPET (T2), and 30 min after CPET (T3). Salivary samples were collected at three time points (T0, T2, T3) using specialized collection kits, and analyses were performed using the ELISA technique. The variations in serum cortisol levels from T0 to T1 did not show statistical significance. Similarly, there was a non-significant decrease in salivary cortisol levels from T0 to T2, with a trend towards significance from T0 to T3. In conclusion, salivary and serum determinations showed significant correlations, indicating that the non-invasive procedure could replace venous blood sampling, although further studies on larger samples are needed [137].

## 5. Hypothalamic–Pituitary–Thyroid Axis

The thyroid gland is extremely important due to its numerous effects. Particularly, its significance lies in its ability to release thyroid hormones (T4 and T3), which are essential for the physiological function of several tissues and organs: thyroid hormones have the capacity to modulate metabolism and act synergistically with other hormones [22,138].

Thyroid hormone synthesis is regulated by feedback mechanisms, both positive and negative, mediated by the HPT axis. A decrease in thyroid hormone levels leads to an increase in thyrotropin-releasing hormone (synthesis of TRH) in the hypothalamus, which, in turn, enhances thyroid-stimulating hormone (TSH) secretion [139,140]. TSH, diurnally secreted in pulsatile mode by the anterior pituitary [141,142], stimulates the thyroid gland to produce thyroxine (T4) and T3, which are stored bound to thyroglobulin (Tg) in large follicles [143]. Subsequently, thyroid hormones T4 and T3 are released into circulation by proteolysis of Tg; T4 is released by the thyroid gland in much greater quantities than T3 (in a ratio of approximately 14:1) [144]. However, T3 is considered the biologically more active thyroid hormone [145].

In healthy humans, about 90% of thyroid hormone is released as T4 and 10% as T3. However, most of the T4 is converted to T3 peripherally by type 1 and type 2 iodothyronine deiodinases (Dio1 and Dio2) [146,147]. Once secreted into plasma, thyroid hormones are mostly bound to plasma proteins (more than 99.7%), such as thyroxine-binding globulin (TBG), thyroxine-binding prealbumin or transthyretin (TBPA), and albumin [148]. Only a small amount of thyroid hormones circulates as a free form: fT4 = 0.03% of total serum T4 and fT3 = 0.3% of total serum T3) [149,150].

Generally, the amount of free thyroid hormone is kept constant through thyroid excretion and release. Variation in TSH and thyroid hormone levels may indicate altered thyroid function. Genetic factors account for up to 65% of interindividual variations in TSH and thyroid hormone levels [151,152], even if many other factors can influence thyroid function. These factors include demographic factors (age and sex [153,154]), intrinsic factors (gut microbiota [155] and stress [156]), medication use [157], and various environmental factors [158,159,160,161].

Thyroid hormone turnover is very slow, making it complex to objectively assess hormonal changes (even relatively large ones) in relation to thyroid gland function [162]. As previously mentioned, the thyroid significantly influences a multitude of tissue–organ functions, as well as growth and development throughout human life, although, this effect diminishes with advancing age [163]. Furthermore, it also has important actions for an individual’s sports performance, which are listed below [164]:-Increased mitochondrial oxidative phosphorylation can lead to an elevation in basal metabolic rate;-Increased tissue response to catecholamines (permissive action), which can have a cardiogenic effect, increasing heart rate and myocardial contractility;-Synergistic effects on growth hormone by enhancing its action;-Facilitation of neuronal maturation process, thus affecting central and peripheral nervous system development and reactivity;-Increased lipid metabolism within skeletal muscle and enhancement of hepatic glycogenolysis, both influencing the glucose turnover rate.

Therefore, thyroid hormones are important energy regulators and can also influence energy processes during sports performance [164,165,166,167,168,169,170,171]. They play a pivotal role in organism adaptation to physical exercise and, therefore, can condition physical performances.

Short-duration, graded physical exercise (≤20 min) results in elevated blood TSH levels as long as a threshold intensity of approximately ≥60% of maximal oxygen uptake (VO_2_max) or above the lactate threshold is reached [77,172]. It is interesting to note that with this elevation of TSH, there is an expected increase in total and free T4, but total and free T3 decrease [172]. The observed hormonal increases mainly appear to be caused by exercise-induced hemoconcentration (i.e., many carrier proteins, such as TBG, are trapped in the vascular space). It is unclear whether the observed reductions are due to decreases in production or increased tissue uptake [77].

The long-term effects of submaximal steady-state exercise (≥60 min) on the thyroid are debatable. Some studies have reported no effect on TSH levels [173], while others indicate that TSH and/or free T3 [174] progressively increase or reach a plateau at approximately 40 min of steady-state exercise [77].

During very prolonged submaximal exercise (~3 h), Berchtold et al. found that total T4 becomes elevated but then decreases post-exercise (i.e., in recovery) [173]. Conversely, in the same study, it was found that total T3 steadily decreased during exercise. Other investigators reported that total T3 remained unchanged, but total T4 increased by 60 min into a prolonged submaximal steady-state exercise session [175]. On the other hand, exhaustive fatiguing endurance exercise only increased circulating free T4 levels [77] (Figure 1). Similarly, fatiguing maximal exhaustive exercise but of shorter duration (graded exercise test) was associated with decreases in TSH and fT4 but increases in total T3 [176] (Figure 1). Finally, repeated low-intensity but demanding physical activity (i.e., field military operations involving sleep deprivation and caloric restrictions) substantially reduce resting T4, T3, and TSH levels, as shown by Opstad et al. [177].

Noticeably, differences in ambient temperature can alter the thyroid response to physical exercise. For example, Deligiannis et al. examined thyroid responses in swimmers exercising in different water temperatures and found that TSH and fT4 were markedly elevated in colder water, unchanged at 26 °C, and decreased in warmer water (T3 levels were unchanged) [178]. This is consistent with others’ work, as cold receptor stimulation regulates changes in TRH and TSH levels [179].

As previously mentioned, thyroid hormones influence growth and development throughout an individual’s life. However, depending on the intensity of the performed physical activity, changes in thyroid hormone metabolism can be observed: generally, children can handle stress stimuli, such as those generated by leisure sports, and the transient increase in stress hormones does not deleteriously affect growth and puberty progression [180,181,182,183,184,185] (Figure 1). Conversely, very strenuous exercise during childhood can negatively influence growth in children and cause delays in skeletal maturation and pubertal progress, sometimes leading to growth potential retention and decreased final height [181,182,183,184,185,186,187,188,189]; in particular, a decrease in thyroid hormones is observed, establishing a condition of hypometabolism. Females seem to be more vulnerable to the detrimental effects of chronic stress and intensive physical training on growth compared to males. Moreover, intensive physical exercise during childhood or early puberty can negatively impact bone acquisition [188].

However, not all children respond uniformly to chronic stress; therefore, the long-term impact of strenuous physical exercise on growth and pubertal maturation may vary among young athletes, thus, continuous and meticulous monitoring of highly trained children is necessary to prevent lasting compromise of their growth potential and pubertal maturation. Typically, stress results in high cortisol which, in turn, reduces T3 but increases rT3.

Under stress, both physical and psychological, such as hunger, an increase in cortisol, which reduces T3 and increases reverse T3 (rT3), a condition known as “low-T3 syndrome”, is observed. The decrease in serum T3 represents an adaptive response of the body to conserve calories and protein, inducing a certain degree of hypothyroidism [190,191] (Figure 1). Higher T4 concentrations might be due to lower circulating T3 levels, via hypothalamic feedback, or to impaired peripheral T4 to T3 conversion (Figure 1) [192], a process that occurs under carbohydrate availability [193]. It has been postulated that LTS may occur when exercise-related energy consumption exceeds calories consumed. However, further studies are needed to clarify the mechanisms influencing the HPT axis during exercise [191].

## 6. Hypothalamic–Pituitary–Gonadal Axis

Acute physical exercise may be associated with an increase, a reduction, or no change in circulating gonadotropin concentrations, depending on the characteristics of the exercise performed. The available data in the literature are not consistent; however, a reduction in LH peaks and suppression of the hypothalamic–pituitary–gonadal axis is commonly observed [40].

Changes in sex hormones in response to exercise occur in both men [194] and women [195], as shown by long-distance running effects on testosterone and amenorrhea, respectively. In women, there is a significant correlation between weekly training mileage and the incidence of amenorrhea.

The most notable endocrine dysfunction linked to exercise training is that which involves disruptions in a woman’s reproductive system, leading to the development of the medical condition known as the female athlete triad. The female athlete triad characterizes a set of conditions seen in physically active young women, encompassing low energy availability (LEA), menstrual irregularities, and reduced bone mineral density (BMD) [196,197,198]. Low energy availability often arises from disordered eating patterns or diagnosed eating disorders, leading to an imbalance between calorie intake and metabolic needs, resulting in an energy deficit [196,197]. While exercise generally yields positive health outcomes for most people, it can become harmful when combined with low energy availability. Several research studies have confirmed that this imbalance disrupts hormonal levels, leading to menstrual irregularities and compromised bone health [198].

Energy availability denotes the quantity of energy that remains and is accessible for bodily functions after deducting the energy expended during daily exercise training from the energy acquired through daily caloric intake from food. Put simply, Energy Availability = Caloric Intake from Diet-Energy Expenditure from Exercise [199].

It is now acknowledged that LEA can lead not only to the triad but also to a condition known as “relative energy deficiency in sports” (REDs). REDs was identified as a distinct entity from the triad by a group of clinicians from the International Olympic Committee [197]. This condition affects both men and women and is characterized by broader impairments in physiological functions, encompassing metabolic rate, menstrual function, bone health, immunity, protein synthesis, and cardiovascular health. These issues arise due to a relative energy deficiency resulting from LEA [197].

In a recent review, Maya et al. [200] observed that hormones play a crucial role in maintaining energy balance through two distinct sets of neurons in the hypothalamus. One set consists of neuropeptide Y/agouti-related protein neurons, which, when activated, promote appetite or food-seeking behavior. The other set comprises pro-opiomelanocortin/cocaine- and amphetamine-related transcript neurons, which, when stimulated, suppress appetite or have an anorexigenic effect. Therefore, energy homeostasis achievement relies on the interaction between the neuronal pathways and energy balance hormones [200].

Ghrelin and leptin are two important hormones that regulate appetite and metabolism. Ghrelin stimulates appetite, and its levels are inversely correlated with body fat, while leptin suppresses appetite, and its levels are directly correlated with body fat. Elite athletes tend to have lower levels of leptin, regardless of menstrual cycle, compared to amateur athletes, which could be an adaptation to maintain their dietary patterns [201]. Additionally, individuals with eating disorders, amenorrhea and anorexia nervosa, exhibit a decrease in ghrelin and an increase in leptin once energy balance is achieved or after weight regain [202,203,204].

These hormonal variations can influence the menstrual cycle in athletes, with higher levels of ghrelin and lower levels of leptin potentially contributing to reduced luteinizing hormone (LH) secretion [205].

Peptide YY (PYY), released by the endocrine L cells of the intestine, increases after meals, signaling satiety. In individuals with a low body weight, such as amenorrhoeic [38,206] athletes and patients with anorexia nervosa [207], PYY levels are higher compared to controls. These elevated levels of PYY are associated with a lower fat mass and body mass index (BMI) and may promote restrictive eating behaviors (Figure 1). Additionally, PYY inhibits osteoblast activity, contributing to lower BMD in athletes and patients with anorexia nervosa (Figure 1).

Oxytocin, a hormone that promotes bone formation, influences metabolism and appetite. Studies indicate that athletes have lower nocturnal levels of oxytocin compared to non-athletes, suggesting a correlation with increased energy expenditure [208]. In amenorrhoeic athletes, lower levels of oxytocin are associated with abnormal bone structure in sites less exposed to mechanical load [208]. Reduced oxytocin levels are also linked to lower resting energy expenditure in athletes, but not in non-athletes, and to lower energy availability in athletes with amenorrhea, implying a role in energy homeostasis under conditions of low energy [209].

The prevalence of menstrual dysfunction in athletes varies depending on the sport, training intensity, and the athlete’s nutritional status. Endurance runners are traditionally at higher risk of functional hypothalamic amenorrhea, associated with lower body weight [210]. A recent study showed runners were more aware of the female athlete triad (Figure 1) compared to dancers and figure skaters, with dancers having twice the risk. Therefore, there is a critical need to educate dancers and other athletes about the triad. Female university athletes in long-distance sports are at a greater risk of stress fractures and bursitis compared to swimmers [211].

Athletes with low energy availability may experience menstrual issues, ranging from irregular cycles to significant absence of menstruation [212]. Amenorrhoeic athletes typically show lower LH pulsatile secretion compared to their eumenorrheic counterparts [205]. However, some studies suggest that oligomenorrheic athletes may exhibit different hormonal patterns, including higher daytime testosterone levels. Overall, hormonal imbalances, such as lower leptin and insulin levels, and higher levels of ghrelin and cortisol, may contribute to suppressing the hypothalamic–pituitary–gonadal axis in states of low energy availability [213,214].

The third factor that characterizes the female athlete triad is an increased fracture risk. Osteopenia and osteoporosis involve reduced bone mass, leading to weaker bones and a higher fracture risk due to inadequate bone development, excessive breakdown, or both. Factors like age of training onset and duration, intensity, and demands of sports influence bone density in athletes [215]. Intrinsic and extrinsic factors contribute to abnormal bone density, with many affected women showing deficiency postmenopause. Estrogen plays a crucial role in bone health by affecting osteoblast/osteoclast activity, inhibiting bone breakdown [216] and affecting other bone-modifying hormones, such as calcitonin, parathyroid hormone (PTH), cytokines, and growth factors [217,218].

Elevated cortisol levels from intense exercise or overtraining increase bone resorption and contribute to bone loss [219] (Figure 1). Leptin levels decrease in response to starvation, leading to lower estradiol levels and amenorrhea, further compounded by low fat storage in athletic women, potentially impacting bone density [219]. An adequate intake of vitamin D (400–800 IU/day) and calcium (1200–1500 mg/day) is necessary for the training period [220].

The impact of exercise training on men’s reproductive endocrinology is not widely recognized by the public. It was previously believed that the male reproductive system was resilient enough to withstand the strain of intense exercise training without consequences. However, current research indicates otherwise. Interestingly, there are several parallels between the reproductive dysfunctions observed in women and men [221].

Hypogonadism, characterized by decreased gonadal function, particularly in males with insufficient testosterone production, can result from abnormalities in the HPG axis [221,222]. Clinical manifestations include a spectrum of symptoms ranging from sexual dysfunction to mood changes and musculoskeletal issues. Testosterone, besides its role in protein turnover and muscle development, also influences erythropoiesis and hemoglobin concentrations, enhancing oxygen transport [221,222,223,224,225,226,227]. These physiological mechanisms are crucial for optimizing athletic performance and adapting to exercise training regimens. Moreover, the presence of low testosterone levels alone, even in the absence of overt symptoms, may indicate “androgen deficiency,” underscoring the importance of comprehensive assessment and management in athletes [221,225,226].

The mechanisms underlying a reduced testosterone/estradiol ratio in OTS are still to be clarified; nevertheless, it is acknowledged that, regardless of the triggers, this process induces a dysfunctional/anti-anabolic response against energy expenditure. According to the Endocrine and Metabolic Responses on Overtraining Syndrome (EROS) study, the testosterone/estradiol ratio should be 13.7:1 (with total testosterone and estradiol expressed in ng/mL and pg/dL, respectively) [228]. Understanding the interplay between hormonal regulation and physical activity is essential for optimizing health outcomes and performance in athletes.

In male athletes, exercise can result in qualitative changes in seminal fluid and infertility. Recent studies have highlighted the negative impact of intense training on sperm morphology and motility in adults with varicocele compared to those without it [229] (Figure 1). While hormonal parameters and physical activity were unaffected, intensive training could exacerbate spermatogenesis issues in athletes with varicocele (Figure 1). Varicocele is a common condition in adolescents, with reported incidences ranging from 9% to 25.8% [230]. Adolescents often engage in high-intensity physical activities, requiring approval for competitive sports from competent authorities. However, there is limited epidemiological research on the effects of intense physical training on varicocele in adolescents [231].

There is growing evidence that certain male athletes may develop a syndrome resembling the female athlete triad. Through scientific studies and clinical investigations, healthcare professionals have gained insights into more effective methods for evaluating, diagnosing, and managing this condition in male athletes, which is now referred to as the male athlete triad [232,233] (Figure 1).

Identifying male athlete triad involves teamwork among healthcare professionals, like sports physicians, dietitians, and mental health experts. Additional members may include athletic trainers, parents (if under 18), medical consultants, and specialists [196,234]. This team assists in diagnosis, treatment decisions, and athlete clearance. Clear and consistent communication among the team members is crucial for maintaining trust and confidence [235].

Several studies have summarized the current symptoms exhibited by athletes, encompassing a thorough assessment of their dietary behaviors, including any fluctuations in weight, current weight goals, and actions taken to manage weight, such as purging or excessive exercise. Moreover, recent illnesses, injuries, or athletic performances were also considered to gain a holistic understanding of their health status [236,237].

The pharmacological history and meticulous documentation of the use of medications that may influence bone health or libido, such as glucocorticoids or serotonin reuptake inhibitors, is also important. Several medications could disrupt the hypothalamic–pituitary–gonadal axis, allowing for a comprehensive understanding of the athlete’s medical background [238]. In addition, exploration of familial medical history and an analysis of the psychosocial factors are important.

## 7. Insulin

Insulin biosynthesis in pancreatic β cells is regulated by numerous mechanisms but is mainly stimulated by the presence of glucose and is increased by cyclic adenosine monophosphate (cAMP). It starts with the synthesis of pre-pro-insulin, which, at the level of the endoplasmic reticulum (ER), undergoes a post-translational modification, becoming proinsulin, which, subsequently, at the level of the Golgi apparatus, undergoes a proteolytic cleavage, producing a signal peptide of 24 amino acids, called peptide C and the insulin matures. Insulin is a peptide hormone with a molecular weight of 5.807 Da, composed of two chains with 21 amino acids (chain A) and 30 amino acids (chain B) connected via two disulfide bridges. After completing biosynthesis, insulin is stored with equimolar amounts of C-peptide in mature granules of the β cells until secretion into the bloodstream [239].

Glucose is the most important physiological regulator of insulin secretion from β pancreatic cells into the bloodstream, along with various factors involved in a complex system necessary for controlling insulin exocytosis [240].

Once secreted by β pancreatic cells, insulin circulates in the bloodstream with an approximate half-life of about 12 min. The insulin receptor is expressed by numerous tissues and organs, and upon its activation, various processes occur [241], some of them particularly critical in elite sports.

Insulin, in concert with other hormones, like glucagon or somatostatin, is the main molecule to control blood glucose levels. Indeed, insulin secretion, triggered by high glucose concentrations (e.g., postprandial) decreases blood glucose levels and inhibits hepatic glucose production [242].

However, the effects of insulin on the entire body are multiple and complex. Insulin promotes GLUT-4 translocation (the glucose transporter found predominantly in skeletal muscle and adipose tissue) from intracellular vesicles to the cell membrane, increasing the glucose rate entry into the target tissue. Excessive glucose transfer to the cells stimulates glycogen formation [243,244], which is critical in endurance sports, where the amount of glycogen stored in muscle acts as an energetic substrate and can affect athletic performance. In addition, protein (muscle) metabolism is significantly affected by catabolic metabolism [245,246] and by insulin, as well [247,248,249,250,251,252]. In fact, based on an insulin-induced anti-catabolic mechanism, protein breakdown is significantly reduced, maintaining as optimal as possible contractile muscle elements.

Acute exercise increases glucose uptake in skeletal muscle through an insulin-independent mechanism [253,254,255,256,257,258,259,260,261]. Acute glucose transport increases following a single bout of body-wide exercise, involving an intramyocellular signaling cascade that includes increased insulin receptor signaling, activation of the AMP-activated protein kinase (AMPK) pathway, phosphorylation of Akt/protein kinase B, nitric oxide production, and calcium-mediated mechanisms involving Ca^2+^/calmodulin-dependent protein kinase (CaMK) and protein kinase C (PKC) [262,263].

An exercise-induced instant effect on glucose occurs primarily through GLUT-4 trafficking [263,264] rather than through insulin receptor signaling, upon the interaction with insulin receptor substrates (IRS)-1, IRS-2, orPI3K [265,266,267]. Since the effects of exercise on insulin sensitivity persist between 16 [268,269] and 48 h [270] after the last exercise, measurements made at these time points in individuals undertaking regular exercise reflect changes in the expression or activity of a variety of signaling proteins involved in the regulation of glucose uptake in skeletal muscle [271].

Insulin-mediated glucose uptake improvements in the whole body after training rely on increased PI3K activity both in rodents [49] and humans [272,273]. These findings are clinically relevant since PI3K activity is decreased in the skeletal muscles of insulin-resistant subjects and patients with type 2 diabetes [274,275,276]. Kirwan et al. [274] reported that insulin-stimulated PI3K activity is higher in skeletal muscles of resistance-trained individuals than in sedentary ones; comparing these two cohorts, PI3K activation correlates with both glucose disposal and whole-body aerobic capacity.

Healthy human skeletal muscles absorb over 80% of an intravenous glucose load [277]; therefore, such an increase in glucose uptake in skeletal muscle after exercise can have a major impact on a whole-body level. Acute exercise increases insulin sensitivity in rat [278,279] and human [280] skeletal muscles.

Metabolic adaptations following physical training cause profound changes in resting metabolism, ameliorating the performance in well-trained individuals, as well. Many of these mechanisms potentially improve insulin sensitivity and improve glucose disposal. Studies comparing trained and untrained subjects have provided insights into the magnitude of this effect, although they do not rule out population-related specific differences. These (mostly initial) studies have shown that trained subjects have a reduced insulin response to a glucose load [281], a lower basal insulin level, and 50% lower insulin levels during constant glucose infusion [282] vs. untrained ones.

Trained individuals were more sensitive to insulin (submaximal insulin stimulation) and showed an equal [283] or increased [284,285] insulin response (maximal insulin stimulation), as detected by a euglycemic clamp. Aging-related decline in insulin sensitivity ameliorates with physical activity, as glycemic and insulin responses in lean and older athletes were similar to younger ones during an oral glucose tolerance test [286].

Training interventions, ranging from few days to several months, in healthy subjects and insulin-resistant subjects (at various insulin resistance degree) provided more direct evidence of physical training-induced benefit on insulin sensitivity.

To quantify insulin-stimulated glucose uptake throughout the body in young subjects after 6 weeks of resistance training, a hyperinsulinemic euglycemic clamp was used [287]. Upon submaximal insulin infusion, glucose absorption increased by 30% after training. Similar results were observed in rats 7 weeks after daily treadmill running [288] and in elderly humans after 16 weeks of strength training [289]. In the latter study, glucose uptake increased by the same amount (±23%) at both physiological and supraphysiological hyperinsulinemia.

The effects of a physical training program in healthy elderly subjects (60–80 years) was investigated by a number of studies [290,291,292]. Long-term schedules (6–12 months) [291] indicate that glucose tolerance remains unchanged in subjects with normal glucose tolerance but with reduced insulin responses and reduced basal insulin levels. The effect is maintained at least 60 h after the last exercise [290]. In the 3-month intervention study of Tonino [292], glucose uptake during a euglycemic clamp was elevated even after 7 days of detraining. Therefore, all these studies suggest a real training-dependent induced effect. In conclusion, it is clear that physical training undeniably increases insulin sensitivity in healthy individuals and reverses aging-associated negative effects on insulin sensitivity. The main effect of training in healthy subjects is that normal glucose tolerance can be maintained with reduced insulin levels.

In the future, the evaluation of hormones, together with the measurements of other biochemical and hematological parameters, should become part of a panel of biomarkers to which athletes, but also people who regularly practice physical activity, should be subjected. The objective is to create an “athlete’s biological passport” to provide all the information on the general state of health. Hormonal responses to exercise are complex, multi-faceted, and affected by several variables. This review could not cover all the aspects associated with exercise-related hormonal responses, and this is one of the limits of this paper, along with the lack of a deeper discussion on each specific consequence on hormone unbalance.

## 8. Conclusions

Intense physical activity is able to modulate the hormonal response and, consequently, all the mechanisms that regulate the body. Hormones play an important role in regulating physiological processes, including energy metabolism, tissue growth, hydration levels, and muscle protein synthesis and degradation. Therefore, it is critical to identify the changes in hormones levels in relation to exercise and understand the physiological functions they influence.

In this scenario, hormone determination and continuous monitoring might represent a valid tool in sports medicine to guarantee safety conditions. Creating ad personam training programs would help maximize performance, maintaining balanced hormone production. We hope that this review contributes to raising awareness on the importance of hormonal modification(s), which have been well known for quite some time but at the same time, at least in part, are still neglected.

## Figures and Tables

**Figure 1 biomolecules-14-01418-f001:**
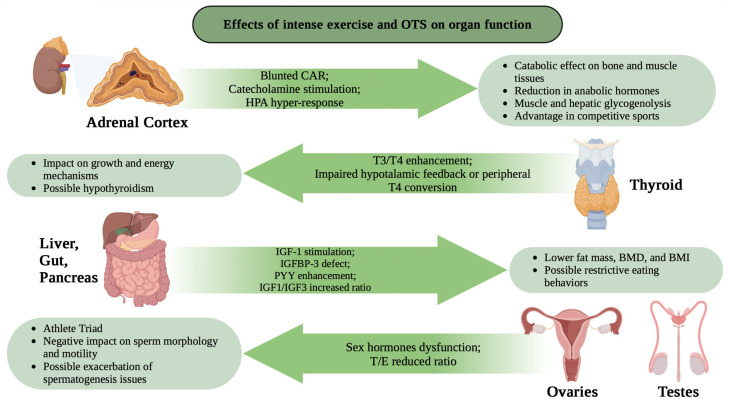
Effects of intense training/OTS on different organ function. The impact of OTS, due to too intense training or insufficient recovery, on some organ functions are summarized. Overtraining syndrome (OTS); cortisol awakening response (CAR); triiodothyronine (T3); thyroxine (T4); insulin-like growth factor-1 (IGF-1); insulin-like growth factor binding protein-3 (IGFBP-3); peptide YY (PYY); bone mineral density (BMD); body mass index (BMI).

**Table 1 biomolecules-14-01418-t001:** General changes in hormone levels induced by physical activity.

	Basal F/M	Training	Acute Physical Exercise	Acute Physical Exercise F/M
GH	↑ F	↑/=/↓	↑	⇑ F
IGF-1	↑ M	↑/=/↓	↑/=	⇑ M
CATECHOLAMINES	F = M	↑/=/↓	↑	↑ M
ACTH	F = M	↑/=/↓	↑	↑ F/= M
CORTISOL	↑ M	↑/=/↓	↑	⇑ M-↑ F
TSH	↑ F/= M	↑/=/↓	↑/=/↓	↑ F/= M
T3–T4	F = M	↑/=/↓	↑/=/↓	F = M
LH-FSH	F = M	=/↓	↑/=/↓	F = M
TESTOSTERONE	↑ M	=/↓	↑/=/↓	⇑ M-↑ F
ESTRADIOL	↑ F	=/↓	↑/=/↓	↑ F
INSULIN	F = M	↓	↓	F = M

Legend: F, female; M, male; ↑, increase; ⇑, substantial increase; =, no variations; ↓, decrease. Growth hormone (GH); insulin-like growth factor-1 (IGF-1); thyroid-stimulating hormone (TSH); triiodothyronine (T3); thyroxine (T4); luteinizing hormone (LH); follicle-stimulating hormone (FSH).

## Data Availability

Not applicable.

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
