# Peer review of "How Does Physical Activity Modulate Hormone Responses?"

_biomolecules, 2024, doi:10.3390/biom14111418_

Round 1

Reviewer 1 Report

Comments and Suggestions for Authors

1) Please be consistent and complete in the use of abbreviations throughout the manuscript.

2) There are extensive references, but the format of the citations is inconsistent, and in error for numerous papers. The authors should take greater care in developing the reference section of the manuscript.

3) At times you make comments that are not in line with the general consensus of thought in the discipline of exercise endocrinology. Please be more careful with your language. I have noted these in the manuscript.

4) The attached PDF file has Adobe "Comments" that list concerns and comments I have. Please address all of them.

Comments on the Quality of English Language

The English is at a high level.

Author Response

We thank this Reviewer for the constructive criticism.

1) Please be consistent and complete in the use of abbreviations throughout the manuscript.

Response:

We have checked and completed the use of abbreviations in the revised text.

2) There are extensive references, but the format of the citations is inconsistent, and in error for numerous papers. The authors should take greater care in developing the reference section of the manuscript.

Response:

We have checked and consistently modified the format of references in the text.

3) At times you make comments that are not in line with the general consensus of thought in the discipline of exercise endocrinology. Please be more careful with your language. I have noted these in the manuscript.

Response:

As you suggested, we have modified the text.

4) The attached PDF file has Adobe "Comments" that list concerns and comments I have. Please address all of them.

Response:

All the suggested comments have been addressed, and all the changes are either highlighted in blue or written in tracked mode in the revised text.

In particular,

Comment 1:  the revised Table 1 indicates sex specific-changes, and abbreviations are explained.

Comment 2: the sentence on cortisol has been modified and addressed according to the R1 suggestion, line 116-118 of the revised manuscript.

Comment 3: Overtraining Syndrome is explained in Figure 1 and  defined, line 132-133, in the revised text.

Comment 4: Growth Hormone is abbreviated in GH after the first definition.

Comment 5: New references have been quoted (79-81).

Comment 6-11: format has been corrected.

Comment 12: the terms “nature” has been changed in “modality”.

Comment 13: Overtraining syndrome is abbreviated in OTS after the first definition.

Comment 14-16: FA has been erased.

Comment 17: all p values in parentheses have been canceled for consistency.

Comment 18: format has been fixed and the first point listed has been changed.

Comment 19: the sentence has been changed.

Comment 20: the correct abbreviation REDs is indicated in the revised text.

Comment 21: citation has been properly corrected.

Comment 22: suggested paper is now quoted.

Comment 23: the list of paper in references has been updated and written in a consistent style; this responds to all the suggestions in reference list.

Comment 24: HERM is addressed in the revised text, line 79-87.

Reviewer 2 Report

Comments and Suggestions for Authors

Firstly, thank you for considering me as a reviewer for this manuscript. The manuscript deals with hormonal response and its influence on physical activity. This topic is precious to the sports community since it can influence athletes' performance.

My comments on the manuscript are listed below:

ABSTRACT

The abstract lacks a conclusion part. Please revise.

TEXT

Line 34 – What is meant by conditioning?

Lines 82-88 – have you considered other factors of sports competition and training on cortisol, such as playing time, contacts in team sports, etc?

Nikolovski, Z., Foretić, N., Vrdoljak, D., Marić, D., & Perić, M. (2023). Comparison between Match and Training Session on Biomarker Responses in Handball Players. Sports, 11(4), 83.

Vrdoljak, D., Gilic, B., Nikolovski, Z., Foretić, N., & Espana-Romero, V. (2024). Hormonal response during official bouldering competition. The Journal of sports medicine and physical fitness.

CONCLUSION

The conclusion section is sparse and short. Consider giving a more detailed conclusion on this topic.

GENERAL

The manuscript is well written and gives high insight into the topic of hormonal changes regarding physical activity.

Author Response

We are grateful to this Reviewer for the comments ameliorating the paper.

ABSTRACT

The abstract lacks a conclusion part. Please revise.

Response:

We have added a sentence for the conclusions in abstract, line 28-30.

TEXT

Line 34 – What is meant by conditioning?

Response:

Conditioning is explained in the revised text, line 36,37.

Lines 82-88 – have you considered other factors of sports competition and training on cortisol, such as playing time, contacts in team sports, etc?

Nikolovski, Z., Foretić, N., Vrdoljak, D., Marić, D., & Perić, M. (2023). Comparison between Match and Training Session on Biomarker Responses in Handball Players. Sports, 11(4), 83.

Vrdoljak, D., Gilic, B., Nikolovski, Z., Foretić, N., & Espana-Romero, V. (2024). Hormonal response during official bouldering competition. The Journal of sports medicine and physical fitness.

Response:

The suggested factors are addressed, line 123-129, and the related paper are quoted in the revised text.

CONCLUSION

The conclusion section is sparse and short. Consider giving a more detailed conclusion on this topic.

Response:

The paragraph conclusions has been modified, according to the suggestions, line 796-810.

Reviewer 3 Report

Comments and Suggestions for Authors

Review report on the manuscript “How does physical activity modulate hormones responses?” (biomolecules-3175385)

Thank you for the opportunity to review this manuscript. The present review focuses on an important topic in the field and appears to be an extended version of an already available online text at https://www.endowiki.it/index.php/news/62-contenuti/patologie-sistemiche (ref 11). There are critical concerns regarding the study design (as a review paper) and the structure of the manuscript. In addition, the novelty of this paper is highly questionable. The following comments may help improve the paper for future submissions.

I understand that this is a narrative review; however, any type of review study should help address knowledge gaps by providing a detailed perspective on the novel aspects of the topic of interest. The present review is neither hypothesis-driven nor does it comprehensively discuss the gaps in available studies focusing on influential variables or a specific population.

Technically speaking, the concept of hormonal responses following PA is complex and multi-faceted. There are various categories of variables (including but not limited to those mentioned in the second sentence of the abstract). However, the present study lacks a systematic discussion on such variables.

Detailed content regarding the study aims and objectives, along with current gaps in the literature, should be clearly explained in the Introduction.

As a critical concern, the design of visual illustrations requires incorporating more advanced techniques using more specific data. Table 1 presents very general directions of changes in hormonal responses, and its degree of informativeness at a scientific level is questionable, as the impact of influential variables has not been elaborated. Figure 1 follows the same approach. It is important to consider that this is a scientific article, not an educational booklet for undergraduate students.

Structurally, the classification of contents and headings does not follow a consistent style.

A standard comprehensive review study is expected to follow established guidelines. However, the present paper falls short of even the minimum methodological expectations for a review study. Even for a narrative review, at least brief explanations about methodological strategies, including “databases and sources,” “primary and secondary variables of interest,” “the procedure of searching,” and “information about pooled data,” are required.

The manuscript has some missing parts e.g., study limitations, practical implications, and suggestions for future investigations which should be added prior to Conclusion.

Regards,

Comments on the Quality of English Language

Almost fine!

Author Response

Thanks to the Reviewer 3 for the comments.

Thank you for the opportunity to review this manuscript. The present review focuses on an important topic in the field and appears to be an extended version of an already available online text at https://www.endowiki.it/index.php/news/62-contenuti/patologie-sistemiche (ref 11). There are critical concerns regarding the study design (as a review paper) and the structure of the manuscript. In addition, the novelty of this paper is highly questionable. The following comments may help improve the paper for future submissions.     

I understand that this is a narrative review; however, any type of review study should help address knowledge gaps by providing a detailed perspective on the novel aspects of the topic of interest. The present review is neither hypothesis-driven nor does it comprehensively discuss the gaps in available studies focusing on influential variables or a specific population.

Technically speaking, the concept of hormonal responses following PA is complex and multi-faceted. There are various categories of variables (including but not limited to those mentioned in the second sentence of the abstract). However, the present study lacks a systematic discussion on such variables.

Response:

We understand the criticism, and we apologize for giving this impression. We intended to summarize the changes of the endocrine system, in order to focus onto hormonal balance/unbalance discriminating health maintenance from disease development. The potential new aspect consists in suggesting the importance to combining different parameters to identify a possible bio-passport to monitor athletes as well as all subjects practicing physical activity. Indeed, we would like to raise awareness on the importance of hormonal modification(s), an aspect well described since quite ago but, in our opinion, still quite neglected, as addressed in the revised text. The limits of the review are addressed in the revised text.

Detailed content regarding the study aims and objectives, along with current gaps in the literature, should be clearly explained in the Introduction.

Response:

We have modified the text, Introduction, line 40-48, and line 161-165.

As a critical concern, the design of visual illustrations requires incorporating more advanced techniques using more specific data. Table 1 presents very general directions of changes in hormonal responses, and its degree of informativeness at a scientific level is questionable, as the impact of influential variables has not been elaborated. Figure 1 follows the same approach. It is important to consider that this is a scientific article, not an educational booklet for undergraduate students.

Response:

Table 1 has been modified and updated with sex-dependent changes; figure 1 has been updated with some effects consequent to exercise-induced changes.

Structurally, the classification of contents and headings does not follow a consistent style.

Response:

We have modified the text.

A standard comprehensive review study is expected to follow established guidelines. However, the present paper falls short of even the minimum methodological expectations for a review study. Even for a narrative review, at least brief explanations about methodological strategies, including “databases and sources,” “primary and secondary variables of interest,” “the procedure of searching,” and “information about pooled data,” are required.

Response:

As this is a narrative review, Methods are not included. However, please find as following the schematic methods used to perform literature research. Please, consider that we could insert this paragraph in the text, if the Reviewer request it.

Methods: PUBMED, SCOPUS, Web of Science and Google Scholar databases were used (no date limit); the terms (hormone OR hormonal OR cortisol OR growth hormone OR HPA axis OR GH-IGF-1 axis OR thyroid OR adrenal cortex OR gonads OR insuline OR cathecolamines) AND (exercise OR physical activity OR sport OR athletes OR overtraining) were used; hand searches of the references of retrieved literature were included and no specific software or digital bibliographic reference manager was used.

The manuscript has some missing parts e.g., study limitations, practical implications, and suggestions for future investigations which should be added prior to Conclusion.

Response:

Following the suggestion, we have modified the text addressing the suggested topics, line 791-798, line 805-815.

All the changes are tracked or highlighted in blue in the revised text.

Round 2

Reviewer 3 Report

Comments and Suggestions for Authors

I have reviewed the revised manuscript and the accompanying response to the first-round comments. While I appreciate the efforts made by the authors to address some of the initial concerns, there remain several critical issues that have not been fully resolved. Scientifically speaking, I regret to inform you that I still do not find the paper suitable for publication in its current form, and my personal decision is still “rejection”.

Comments on the Quality of English Language

--